# Microfluidic Manipulation for Biomedical Applications in the Central and Peripheral Nervous Systems

**DOI:** 10.3390/pharmaceutics15010210

**Published:** 2023-01-06

**Authors:** Zhenghang Li, Zhenmin Jiang, Laijin Lu, Yang Liu

**Affiliations:** Department of Hand Surgery, Center of Orthopedics, The First Hospital of Jilin University, Changchun 130021, China

**Keywords:** microfluidic platforms, nerve injury repair, damage models, tissue engineering scaffold, drug-delivery system

## Abstract

Physical injuries and neurodegenerative diseases often lead to irreversible damage to the organizational structure of the central nervous system (CNS) and peripheral nervous system (PNS), culminating in physiological malfunctions. Investigating these complex and diverse biological processes at the macro and micro levels will help to identify the cellular and molecular mechanisms associated with nerve degeneration and regeneration, thereby providing new options for the development of new therapeutic strategies for the functional recovery of the nervous system. Due to their distinct advantages, modern microfluidic platforms have significant potential for high-throughput cell and organoid cultures in vitro, the synthesis of a variety of tissue engineering scaffolds and drug carriers, and observing the delivery of drugs at the desired speed to the desired location in real time. In this review, we first introduce the types of nerve damage and the repair mechanisms of the CNS and PNS; then, we summarize the development of microfluidic platforms and their application in drug carriers. We also describe a variety of damage models, tissue engineering scaffolds, and drug carriers for nerve injury repair based on the application of microfluidic platforms. Finally, we discuss remaining challenges and future perspectives with regard to the promotion of nerve injury repair based on engineered microfluidic platform technology.

## 1. Introduction

The nervous system of organized vertebrates plays a pivotal role in maintaining the close connection between its internal structures and the external surroundings [1]. This system is a highly complex network that controls and regulates the activities of all other systems and is categorized into the central nervous system (CNS) and peripheral nervous system (PNS). For PNS injury, direct end-to-end repair without tension is the preferred surgical method. However, for direct epineural repair with excessive tension, autologous nerve grafts remain the gold standard for bridging nerve defects [2,3]. Disorders of the PNS frequently have overwhelming and disturbing impacts on the daily life and habits of patients, although these disorders are not generally considered life-threatening [1]. Physical injuries and neurodegenerative diseases often lead to irreversible damage and dysfunction of the CNS, even inducing permanent disabilities [4]. Most axons of the adult mammalian CNS fail to regrow and achieve functional recovery after injury, whereas axons of the mammalian PNS can spontaneously regenerate, at least to a certain extent [5]. In mammals, successful axonal regeneration requires not only a good intrinsic growth capacity of the damaged neurons but also a permissive extracellular environment [6]. A major challenge in promoting functional recovery following injury and disorders of the adult mammalian CNS and PNS has been the ability to identify the intrinsic mechanisms that orchestrate a regenerative response and enable long-distance axonal regeneration. Although traditional cell culture techniques have made important contributions to our understanding of nerve regeneration, they are unable to accurately manipulate the behavior of neuronal cells and regulate the extracellular environment [7].

Advanced microfluidic technology can provide us with the ability to explore the nervous system at the micro level in a highly precise manner. Over recent years, microfluidic technology has been increasingly applied in the construction of neural system models and nerve injury repair scaffolds [8,9]. Microfluidics technologies provide us with rapidly developing and versatile technologies for studying integrated neural cell-sized microarchitecture [10,11]. Understanding this microarchitecture could provide us with the ability to control axonal behaviors and regulate the extrinsic microenvironment by the application of micro-scale methods [12]. In addition, the combination of microstructural and biological factors through a microfluidic platform could further simulate the microenvironment in vivo, which could enable us to explore the corresponding structure and function of nerve cells or organoids [8,13]. Currently, the simulation of different types of nerve damage and the establishment of a microenvironment for nerve regeneration based on microfluidic platforms are the main in vitro models used for investigating neurodegeneration and regeneration. Therefore, these in vitro models based on microfluidic platforms can provide an efficient and important tool with which to investigate relevant biological mechanisms and provide a theoretical basis for the preparation of regenerative scaffolds and drug delivery for nerve injury.

In this review, we sought to provide an overview of the basic biological mechanisms that control axonal regeneration after nerve damage and focus on the tissue engineering scaffolds and drug carriers used by microfluidic devices and applied in models of nerve injury and the neuroregenerative microenvironment. Having multiple in vitro model systems, tissue engineering materials, and drug-delivery preparations based on microfluidic platforms could enhance the understanding of the complex mechanisms of nerve injury and regeneration, thus providing new therapeutic strategies for solving clinically relevant problems.

## 2. Neurological Injury and Disorders of the CNS and PNS

### 2.1. Basic Anatomical Structure and Functionality of the Nervous System

As the most important functional regulation center of the human body, the nervous system is the most complex system and consists of billions of neurons and glial cells. The nervous system plays a leading role in the body, controlling and regulating the activities of all other systems and thus making the human body an organic entity. The nervous system acts as a unit in terms of structure and function and is categorized into the CNS and PNS. The CNS is composed of the brain and spinal cord and serves as the highest-level control center of the human body by receiving, interpreting, and conducting neuronal signals [14]. In contrast, the PNS is composed of various nerves arising from the CNS, including the cranial, spinal, and visceral nerves, and is responsible for transmitting signals between the CNS and other parts of the body [15,16]. In the PNS, impulses in the sensory nerves are transmitted from specific receptors to the center by afferent nerves, while impulses in the motor nerves are transmitted from the center to the periphery by efferent nerves. Neurons in the PNS and CNS represent a highly polarized cell population, and axonal terminals are usually located far away from the cell body [17]. This unique morphology is essential for neurons to be able to perform normal functions. 

Glial cells are widely distributed in the CNS and PNS and are recognized as functional and structural components of the nervous system [18]. These cells play a role in the development of the nervous system and homeostasis, including supporting and separating the neurons, participating in the immune responses, and the formation of the myelin sheath and the blood–brain barrier (BBB) [19]. Glial cells in the CNS consist of astrocytes, oligodendrocytes, and microglia, while Schwann cells are the primary multifunctional glial cells of the PNS. In the CNS, oligodendrocytes myelinate neuronal axons to increase conduction velocity and provide metabolic support; the dynamic regulation of myelination may adjust the precise timing of information communication across functional circuits [20,21]. However, Schwann cells are responsible for the myelination of axons in the PNS and are surrounded by the basal lamina, which is a contiguous and longitudinal structure throughout the axon. The presence of the basal lamina is an important feature that distinguishes the PNS from the CNS and may promote axonal growth and shield axons from inhibitory molecules during neural regeneration [22]. Significantly, the lack of a basal lamina might be a critical factor responsible for regenerative failure in the CNS. 

Another important distinction between the CNS and PNS is the number of myelinated axonal segments per myelinating cell (up to 60:1 versus 1:1) [23]. Axonal transport is considered a pivotal process in nerve injury and repair; fast axonal transport can deliver a cargo in antegrade (away from the cell body) and retrograde (toward the cell body) directions, while slow axonal transport can only deliver cytoskeletal proteins in an antegrade direction [24]. Nerve fibers in the PNS act as a functional unit and are composed of an axon and associated Schwann cells. Each individual nerve fiber is protected by the endoneurium (mainly collagen and elastic elements) along with the endoneurium enveloping nerve fiber groups into nerve fascicles which are wrapped by the perineurium (mainly connective tissue) [25]. Furthermore, several fascicles are held together into a nerve trunk by the epineurium (composed of mainly irregular connective tissue and adipose tissue) [24]. The complex microcirculation in the PNS runs along the nerve longitudinally and provides oxygen and a nutritional supply [26]. The endothelial cells in the endoneurial microvessels form the blood–nerve barrier (BNB) to maintain endoneurial homeostasis [25]. However, due to the absence of astrocytes, this physical barrier is considered to be looser than the BBB in the CNS [27,28].

### 2.2. Axon Regeneration in the Peripheral Nervous System

Peripheral nerve injuries are a common affliction in the clinic and are associated with a variety of symptoms and signs depending on the nerves involved and the severity of damage [29]. Axonal regeneration and functional recovery in the PNS can often occur following injury because peripheral neurons possess remarkable regenerative abilities after injury. However, severe damage to the peripheral nerves may also lead to permanent and irreversible neurological deficits, including the motor dysfunction of the target organs and the development of refractory chronic pain [30]. In these cases, even if nerve grafts can provide a bridge to promote axon regeneration, functional recovery is always slow and limited [26,31]. The regenerative ability of the PNS is affected by extrinsic and intrinsic factors, which create a growth-permissive microenvironment where the activation of the intrinsic cells leads to successful axonal regeneration [32]. Therefore, to ultimately improve the degree of functional recovery after nerve injury, we need to consider a variety of strategies to explore and understand the intrinsic mechanisms underlying the regeneration of peripheral neuronal axons. 

In the PNS, complex cellular and biochemical responses can occur in the neurons, axons, and non-neuronal cells at the specific site of injury. The rupture of the axonal membrane after nerve injury, combined with the inversion of the calcium/sodium exchange pumps, results in a rapid influx of calcium into the axoplasm, and this initiates the generation of high-frequency action potentials along the proximal axonal region to the soma [33,34]. Calcium influx can govern axon regeneration by triggering various cellular autonomous mechanisms that are required for successful axonal growth, including the regulation of intracellular pathways and epigenetic changes [32]. Importantly, this reverse calcium wave invades the cell body and causes the activation of protein kinase Cµ (PKCµ) and the subsequent nuclear export of histone deacetylase 5 (HDAC5), thus increasing the levels of histone acetylation and activating the pro-regenerative transcription program [35]. Furthermore, calcium influx activates cyclic adenosine monophosphate (cAMP) and protein kinase A (PKA), thus signaling to DLK-1, resealing the axonal membrane, and promoting local protein synthesis and growth cone formation [36,37]. The process of rebuilding a functional growth cone in the axon tip is characterized by the organization of the microtubule cytoskeleton and mitochondrial transport [38,39]. This process also involves the activation and retrograde transport of other injury signals, including extracellular signal-regulated kinase (ERK), c-Jun N-terminal kinase (JNK), and signal transducer and activator of transcription 3 (STAT3) [40,41,42]. In the cell body, regeneration-associated genes (RAGs) are expressed and transported in an anterograde manner to mount a regenerative response, including arginase-1 (Arg1), growth-associated protein-43 (GAP-43), and interleukin-6 (IL-6) [43,44,45]. This series of gene-expression programs supports the formation of dynamic growth cones and the subsequent axonal extension required for peripheral nerve regeneration.

When the neuronal cell body and proximal axon are ready to grow, the distal axon will also undergo a series of cellular and molecular changes. This progressive breakdown of the axonal cytoskeleton and clearance of distal fibers is known as Wallerian degeneration, a process that creates a suitable microenvironment for axonal regrowth and reinnervation [46,47]. Schwann cells play a crucial role during axonal regeneration (Figure 1) [16]. In the PNS of adults, Schwann cells are highly specialized and quiescent glial cells. However, following injury, and in response to the ERK signaling pathway, Schwann cells are prompted to de-differentiate and proliferate to serve as central mediators of regeneration [48]. These repaired Schwann cells execute a regenerative program that involves controlling the breakdown of the BNB and the subsequent infiltration of inflammatory cells, myelin autophagy, and the activation of neurotrophic factor expression [48,49,50]. Eventually, their functionality causes the formation of Büngner bands, guiding the growth cone across the tissue bridge that connects the proximal and distal nerve stumps [51].

Moreover, successful PNS regeneration also relies on the coordinated contribution of immune cells [52]. The combined action of Schwann cells and macrophages clears the myelin debris after Wallerian degeneration and creates a permissive microenvironment for regenerating axons to regrow [53,54]. Therefore, inflammation is recognized as an important aspect of nerve regeneration, and its extent and duration can affect the final outcome of regeneration [55]. In addition to the dominant role of hematogenous macrophages, resident macrophages are also able to induce inflammation and play a role in the early phagocytosis of myelin [56]. Anti-inflammatory phenotype (M2) macrophages can respond to injury-induced hypoxia and increase the expression of VEGF-A (vascular endothelial growth factor A), thus leading to the proliferation and migration of endothelial cells and the formation of Büngner bands [57]. Indeed, macrophages exhibit a pivotal role during peripheral nerve regeneration. The rate of peripheral nerve regeneration in humans depends on slow axonal transport, occurs at approximately 1–3 mm per day, and may further diminish over time [46]. In addition, the misdirection of regenerative axons is another important factor that can lead to poor functional reinnervation of the target organs [58]. Although peripheral nerve injury is not life-threatening, it always results in limb disability and leads to a notable decline in a patient’s quality of life [59]. Future research should focus on the intrinsic mechanisms responsible for peripheral nerve regeneration, including the gap distance, regeneration rate, Wallerian degeneration, regeneration specificity, and end-organ viability.

### 2.3. Axon Regeneration in the CNS

Physical injuries and neurodegenerative diseases often lead to irreversible damage and dysfunction in the CNS and may even induce permanent disabilities [4]. Although neurons in the CNS exhibit a robust elongation capacity during development, this spontaneous growth capacity dramatically declines after parturition [60]. In addition, inhibitory factors in a non-permissive tissue environment are also involved in axonal regeneration, including the glial scar, myelin debris, and axonal components [61,62]. Therefore, in contrast to the PNS, regenerative failure is a characteristic manifestation of the damaged CNS neurons in adult mammals. In contrast, a conditioning PNS lesion has been described as a classical paradigm for stimulating axonal regeneration in the CNS: a prior injury to the peripheral axons [5]. This positive promoting effect may arise from the activation of the regenerative program prior to CNS damage [32]. The conditioning lesion can trigger specific transcription factors, activate the expression of RAGs, and increase the synthesis of cytoskeletal elements and the transport of mitochondria [63,64]. Although we have already discovered some aspects of regenerative signaling, there is little known about the cellular biological processes and basic intricate mechanisms that are regulated by the conditioning lesion to promote axonal regeneration. 

In the CNS, the injured axons often form a retraction bulb at the lesion site, and this non-growing growth cone exhibits disorganized microtubules. Indeed, pharmacological manipulation of the microtubule cytoskeleton can interfere with the formation of retraction bulbs and promote regeneration and functional recovery in the CNS [65,66]. The activation of the pro-regenerative program after injury may be hindered by five key aspects in the CNS (Figure 2). First, the reduced concentrations of calcium concentrations in the injured axons may lead to the failure of resealing in the broken membrane, thus causing nerve degeneration [32,67]. Second, an increase in histone acetylation cannot induce axonal regeneration following axonal injury [35]. Third, the lack of robust RAG expression also results in a restricted potential to regenerate [68]. Fourth, the capacity to synthesize new proteins is limited at the injury site, thus resulting in axonal regenerative failure [69]. Finally, the signaling pathways and transcription factors involved in axonal regeneration are downregulated by certain inhibitors of axonal regrowth. For example, the inhibitory effect of phosphatase and tensin homolog (PTEN) can cause the levels of the mammalian target of rapamycin (mTOR) to decline, thus influencing the regenerative potential of CNS neurons in adults [70]; the suppressor of cytokine signaling (SOCS) proteins can inhibit Janus kinase (JAK) and STAT activation, thus creating a negative feedback signal that reduces pro-regenerative signaling involving cytokines and growth factors [71]; and the exchange factor for ADP-ribosylation factor 6 (EFA-6) is also known as an intrinsic regulator of regenerative capacity and can inhibit axonal regeneration by interfering with the transport of endosomal integrin [72].

Major extrinsic factors that regulate CNS regeneration are associated with myelin debris from the injured axons and the formation of glial scars. Compared to the PNS, CNS injury is obviously associated with the delayed clearance of myelin debris due to the ineffective phagocytosis of oligodendrocytes and microglia, thus forming an obstacle to axonal regeneration through the injury site [73]. Oligodendrocytes, the CNS equivalent to Schwann cells, lack pro-regenerative abilities and can exhibit multiple anti-regenerative effects (Figure 1). Foremost, oligodendrocytes play a vital role in the formation of glial scars, which is considered another impediment to CNS regeneration. Glial scars are primarily composed of reactive astrocytes and chondroitin sulfate proteoglycans (CSPGs), as well as semaphorin 3A and ephrin B [74,75,76]. CSPGs are secreted by astrocytes and oligodendrocytes and can exert a range of anti-regenerative effects, including the inhibition of axonal regrowth, the restriction of neuronal plasticity, the masking of growth-promoting proteins and the transformation of growth-attracting proteins into repulsive signals [16]. In addition, oligodendrocytes can also secrete inhibitory factors that prevent nerve regeneration, such as Ephrin-B3, semaphorin 5A, semaphorin 4D, and netrin-1 [77]. Interestingly, astrocytes and non-astrocytic cells within glial scars also express multiple axon-growth supporting cues [78]. Therefore, glial scars should be viewed as a complex environment involving both inhibitory and permissive signals that can influence the extension of regenerative axons through the site of injury [79]. In the CNS, microglia and macrophages exhibit a dichotomous effect on regenerating axons, thus releasing both pro- and anti-inflammatory cytokines [16]. Importantly, microglia can induce the propagation of secondary degeneration following the initial insult and participate in the formation of glial scars, thus significantly inhibiting axonal regeneration [80]. On the other hand, microglia can phagocytize myelin debris and protect against glutamate excitotoxicity, thus generating a pro-regenerative effect. Collectively, these findings show that microglia can serve as both glial cells and immune cells, not only leading to progressive degeneration but also controlling inflammatory processes and neuronal regeneration [81]. Therefore, gaining a better understanding of immune responses in the CNS is crucial if we are to develop effective therapies for improving neural regeneration.

To overcome regenerative failure in the CNS, multi-faceted experimental approaches should be conducted to further understand the basic cellular biological processes responsible for axonal regeneration. Furthermore, strategies should be implemented to manipulate intrinsic regenerative mechanisms and form the basis of future regenerative treatment after CNS damage.

## 3. Microfluidic Platforms for Exploring Nerve Injury and Regeneration

Traumatic injuries or neurogenerative diseases often lead to irreversible damage and dysfunction of the PNS/CNS and may even induce permanent disabilities. Though we have conducted in-depth research investigating the process of neuronal degeneration and regeneration, the key mechanisms that regulate axonal regeneration have yet to be elucidated. Excitingly, various microfluidic approaches have been used to explore the mechanisms of nerve injury and regeneration (Table 1).

### 3.1. Engineering Neural Damage Using Microfluidics

Various in vitro models need to be implemented to explore the intrinsic mechanisms of nerve injury and repair, and such research will serve as the basis of future regenerative treatments after nerve damage. Compared to in vivo animal models, in vitro biological experiments based on a microfluidics platform are rapid and efficient and are widely used in disease modeling [105,106,107,108,109]. In vitro models constructed using microfluidic technology can allow for the controlled manipulation and targeting of neurons or axons and also provide a means of simultaneously monitoring their behavioral changes in real time. Axons can be successfully isolated from cell bodies by deploying microfluidic devices through compartmentalization; this is a vital strategy that can be used to identify specific neural regenerative mechanisms [110,111]. Microfluidic platforms have been extensively used for sorting and classifying neuronal cells, axonal guidance, isolating dendrites and synapses, engineering the neural microenvironment, and engineering neuronal networks [111,112,113,114,115,116]. To date, it is still difficult to explore the cellular and molecular mechanisms involved in neural injury and neurodegenerative diseases. Different neural damage models can be built by using microfluidic technology; this strategy can provide a means for studying the mechanisms associated with nerve regeneration.

#### 3.1.1. Traumatic Neural Injury Models

Traumatic injuries of the PNS and CNS induced by chemicals are often encountered clinically. Microfluidic devices have allowed for the establishment of chemical injury models of axons. For example, Yang et al. [82] utilized microfluidic channels to demonstrate the toxic effects of paclitaxel on axons (Figure 3A). These authors also identified the significant axonal degenerative effect of paclitaxel when applied to the axonal side using compartmentalization. In addition, human erythropoietin has been shown to be neuroprotective against this type of chemotherapy-induced neurotoxicity. This observation facilitated the understanding of axonal degeneration and neuroprotection and was also of great significance for the development of neuroprotective drugs for peripheral neuropathies. In another study, Li et al. [83] provided an integrated microfluidic device to investigate neuronal degeneration after localized chemical toxin stimulation and subsequent regeneration following a co-culture with desired cells in a spatiotemporally controlled chamber or treatment with the drug monosialoganglioside. The results of this previous study demonstrated that axons were more resistant to acrylamide injury than the soma because the somal injury could generate secondary axonal collapse. This study established a foundation for future controlled and multifactor neural compartment regeneration after nerve injuries. 

Mechanical damage is also an important and common form of neural injury. Microfluidic platforms have provided various in vitro models for mechanical injuries in axons using aspiration, strain, and laser ablation methods [84,85,86,117]. Dolleé et al. [85] used a brain-on-a-chip microfluidic approach to investigate the neuronal response to a mechanical diffuse axonal injury (Figure 3B). The authors applied pressure through a cavity to stretch axons to varied strains by manipulating the dimensions of the microchannels to mimic diffuse axonal injury in traumatic brain injury; ultimately, they showed that the increased axonal injury was proportional to the extent of the increased strain and concluded that the diameter of the axons plays a crucial role in strain injury. In addition, the authors also demonstrated that mitochondrial health had significant effects on axonal degeneration, which deepened our understanding of the biochemical results of axonal injury and revealed that ethylisopropyl amiloride (EIPA), a sodium–hydrogen exchange inhibitor, could be used as a potential therapeutic for axonal strain injury. In another study, Kim et al. [86] developed a neuro-optical microfluidic device for investigating axonal injury and subsequent regeneration (Figure 3C). This platform could allow for a femtosecond laser to produce precise axotomy and enable the continuous long-term observation of post-injury events occurring from initial degeneration to subsequent regeneration, facilitating an understanding of the neuronal response to selective induced axonal injuries. Thus, this platform was considered a practical device for studying the neuronal response to mechanical injury.

#### 3.1.2. Neurodegenerative Disease Models

Neurodegenerative diseases are common in elderly patients and can both seriously affect the quality of life of patients and bring a serious burden to their families and society [118,119,120,121]. At present, most neurodegenerative diseases are still incurable, and the specific neurobiological mechanisms controlling the occurrence, development, and treatment of these diseases remain uncertain. Therefore, the preparation of in vitro neural damage models is of great value for studying the intrinsic mechanisms underlying neurodegenerative diseases. Alzheimer’s disease (AD) is a complex and multifactorial neurodegenerative disease and has become the major cause of dementia. In a previous study, Shin et al. [87] presented a physiologically relevant three-dimensional (3D) microfluidic-based model which incorporated a brain endothelial cell monolayer with properties similar to those of the BBB, thus successfully establishing an AD culture system. In another study, Park et al. [88] developed a 3D tri-culture microfluidic platform to model human AD in vitro by adding inflammatory activity, thus mirroring the microglial recruitment and neurotoxic activities that cause damage to AD neurons and astrocytes (Figure 3D). The success of this model fostered the understanding of the molecular mechanisms of AD, and it could also be used to screen therapies for stopping or reversing neurodegeneration in AD. In addition, this in vitro model has enabled the development of more suitable and reasonable human AD models for further relevant mechanistic studies. 

Parkinson’s disease mainly affects dopamine-producing neurons in the substantia nigra and is considered the second-most common neurodegenerative disorder [120,122]. In a previous study, Moreno et al. [89] proposed an in vitro model of Parkinson’s disease using a 3D microfluidic cell culture system that differentiated neuroepithelial stem cells into dopaminergic neurons (Figure 3E). This new advanced 3D in vitro model integrated the innovations of developmental biology and microfluidic cell culture and could be used to personalize drug discovery for Parkinson’s disease. Amyotrophic lateral sclerosis (ALS) is a fatal neurodegenerative disease of the CNS that involves the progressive degeneration of motor neurons. At this time, this disease remains rare but is difficult to recognize and diagnose [121]. In a previous study, Osaki et al. [90] developed a novel ALS-on-chip model from 3D skeletal muscle bundles and motor neurons using a microfluidic device, which could help to identify the precise pathogenesis of ALS. 

### 3.2. Engineering Neural Microenvironment Using Microfluidics

Unlike other cells in the human body, neurons form highly specific tissue structures with unique morphological and electrophysiological characteristics, thus making the nervous system one of the most sophisticated systems in the human body. Compared to traditional research methods, microfluidics can operate neurological cells and tissues more accurately at the micro level, such as drug delivery [123], cell reprogramming, and transfection. Microfluidics with compartmentalization is a feasible method with which to deal with neuronal soma and axons. In traditional research efforts, in vivo experiments cannot readily explain the complex mechanisms underlying nerve injuries and neurodegenerative diseases due to the complex nerve structures and interactions between not only cells and tissues but also the cells themselves. The use of in vitro models with microfluidics technology can simplify the environment of neurological cells while ensuring the provision of appropriate microenvironment conditions. In terms of simulating the microenvironment of stem cells and Schwann cells, microfluidics highlight key factors and weaken irrelevant effects, thus making the data produced clearer and more credible. The application of microfluidics in engineering the neural microenvironment has become an inevitable trend for investigating injuries and diseases in the nervous system [124].

#### 3.2.1. Engineering the Neural Microenvironment to Differentiate and Grow

After nerve injuries, axons must bypass substrates that do not allow for axon growth, and some growth factors and other facts such as relevant molecules may help to guide regeneration [125]. In the mature nervous system, certain molecules, such as neurotrophic factors, are secreted by cells and control synaptic function and synaptic plasticity while continuing to regulate the survival of neurons. On the one hand, these factors play an important role in the development of the nervous system. It would be beneficial to establish a gradient of multiple positive factors by microfluidic equipment to play a guiding role in axonal regeneration [126,127]. On the other hand, the directional induction of stem cells has been widely used in research targeting neural tissue repair and has demonstrated strong value for repair. 

In their study, Kothapalli et al. [91] applied a microfluidic device to study axonal guidance under a chemical gradient. Hippocampal or dorsal root ganglion (DRG) neurons were implanted into microchannels, and the encapsulated cells were then seeded onto the surface of a 3D collagen gel (Figure 4A). In the matrix, neurites grew inward in a manner perpendicular to the gradient of chemical clues. Collectively, these results showed that microfluidic devices can produce stable chemical gradients that may be used to study the co-culture of neural stem cells (NSCs) and other cell lines such as glial cells. Furthermore, this technology can be used to study the neurobiology, cell migration, and matrix remodeling and has potential application value for tissue engineering and regenerative medicine. Recently, Tolomeo et al. [92] used a messenger RNA (mRNA)-based microfluidic system to induce pluripotent stem cells (PSCs) to differentiate in different directions; furthermore, NGN2 mRNA was found to induce PSCs to differentiate into neuronal cell lines (Figure 4B). These results demonstrated that this system facilitated the evaluation of mRNA-based transcriptional programming applications and also had the potential to determine the specific contribution of each transcription factor in a desired transcriptional programming trajectory.

The inherent characteristics of microfluidic equipment make it possible to precisely control the time and space of the cell culture microenvironment, thus making it possible for cells to develop into tissues similar to the structure of the body [128]. One previous study introduced a gradient-generation microfluidic platform to optimize the proliferation and differentiation of NSCs [93]. Microfluidic equipment prepared from poly(dimethylsiloxane) elastomers (PDMS) was used to culture human NSCs (hNSCs) in a continuous flow of a growth factor concentration gradient that minimized autocrine and paracrine effects. Growth factors, such as epidermal growth factor (EGF) and fibroblast growth factor 2 (FGF2), were included in the concentration gradient. Finally, it was revealed by microscopy and immunocytochemistry that the hNSCs had proliferated and differentiated into astrocytes, with both their proliferation and differentiation varying in line with the concentration of the growth factor (GF) in terms of both grading and proportion. Some researchers envisage using bioengineered stem cells to replace lost or diseased central and peripheral neurons and/or Schwann cells in the PNS. These authors hope that differentiated stem cells can assume the dual role of neurons and Schwann cells and promote Schwann cells to form neuronal myelin sheath tissue. Along these lines, Ramamurthy et al. [94] used ultra-slow-flow microfluidic equipment to form a gradient flow containing nutrients, growth factors, or neurotrophic cytokines, in which stem cells differentiated into neuron-like cells and Schwann cell-like cells (Figure 4C). Their results showed that differentiated Schwann cells could form the myelin sheath of neurons in vitro. Furthermore, both populations of induced cells could provide viable therapeutic approaches to solving the problem of sensorineural hearing loss.

Elsewhere, Park et al. [95] presented a novel neuronal co-culture microsystem with multi-compartments, including one somatic compartment for culturing neurons and six axon/glial compartments for culturing oligodendrocytes (OLs) (Figure 4D). The somatic and axonal/glial compartments were connected by an array of axon-guiding microchannels that acted as physical barriers, thus confining neuronal cell bodies within the somatic compartment while allowing for axonal growth into the axon/glial compartment. Microchannels also enabled fluid isolation between compartments, thus allowing for six independent experiments on a single device for higher-throughput CNS axon-glia interaction studies in vitro. Therefore, this microsystem can be used to study the signaling networks of the CNS and screen the growth factors or drug candidates that promote myelin repair.

#### 3.2.2. Engineering Neuronal Networks and Organoids 

To establish a microscopic model of neural tissue that provides similar circumstances as those in vivo, it is necessary to consider not only the physiological environment of cells but also the morphology and function of tissues. In vivo, a single neuron is embodied into a larger and more sophisticated longitudinal tissue structure, thus forming a bundle of axons or a nerve. Kitagawa et al. prepared an alginate saline gel microfiber by applying a multi-layer microfluidic device with a structure composed of micro-nozzle arrays (Figure 5A) [99]. Neuron-like PC12 cells were wrapped in a parallel area composed of a soft water gel matrix to form grooved fibers. After two weeks of culture, a millimeter-long intercellular network was formed, which mimicked the structure of complex nerve bundles in the body. This fiber could be used to simulate other linear structures in the human body and is suitable for the preparation of nerve network models. Therefore, microfibers have the potential to investigate the developmental dynamics and differentiation behaviors of neuronal cells and to support the development of new therapeutic drugs for neurodegenerative diseases.

Compared to a two-dimensional neural network, neural network models with a 3D structure are more similar to the physiological environment in the body and possess many advantages. In a manner different from other 3D models, organoids of the brain show structural characteristics of the developing brain [129,130]. It is still challenging to conduct bionic engineering for the brain in a manner that is conducive for brain development. Organoids based on microfluidic platforms can also be applied for microenvironment simulation, disease modeling, and drug screening. In Zhou et al.’s work [104], a printing technology supported by a lipid-bilayer was developed to 3D-print human cortical cells in the biocompatible ECM (Figure 5B). The printed hNSCs showed neural differentiation and the formation of a functional neural network. Therefore, precise pre-patterning by 3D printing can produce natural and unnatural structural patterns, which can be applied to study human cerebral cortex developmental processes. Wang et al. [96] used an organ-on-a-chip system to generate 3D brain organoids derived from hiPSCs in vitro (Figure 5C). Their system provided a similar microenvironment to that of the brain by integrating 3D Matrixgel, fluid flow and multicellular architectures of tissues. Under perfused culture conditions, the prepared brain organoids showed neuronal differentiation, brain regionalization, and cortical spatial organization, thus successfully modeling early human brain development. This simple and powerful brain organ-on-a-chip technique may further improve the maturity of various stem cell-derived organoid engineering efforts in the nervous system. 

In complex neural networks, the normal structure of the BBB is very important for drug screening and microenvironment homeostasis. Noorani et al. [97] developed a new BBB model in vitro by using microfluidic-based organ-on-a-chip technology and mimicked the hemodynamic and structural characteristics of cerebral microvessels that could not be realized using traditional two-dimensional platforms. Their study showed that a multicellular co-culture system had great potential for the study of CNS diseases. In another study, a microfluidic BBB model was constructed with human cerebral microvascular endothelial cells (hCMEC/D3) to evaluate the efficiency of sunitinib passing through the BBB (Figure 5D) [98]. They confirmed the fascinating advantage of this platform, which can simultaneously estimate the BBB permeability and anti-tumor activity of drug candidates for treating CNS diseases.

#### 3.2.3. Engineering Drug Delivery for Neural Tissue

Drug-delivery carriers can be engineered using microfluidic platforms and have specific shapes, sizes, and monodispersities [131,132]. Droplet microfluidics, also referred to as segmented flow microfluidics, is an important aspect of microfluidics technology that has been extensively investigated [133,134,135]. Segmented flow microfluidics can generate drug carriers and achieve the high-throughput filtering of drugs by using droplets as small vessels and formers. Droplet microfluidics enable the precise control of the fluid moving into the microchannels, thus resulting in droplets and particles with various configurations and morphologies. The simplest system of droplet microfluidics is a single emulsification, which can be used to produce microspheres. The use of microspheres loaded with neural therapeutic drugs can make local neural treatment more accurate [136]. Wei et al. [100] designed a core/shell-structure microcomposite using microfluidic technology, which consisted of a polymer core containing methylprednisolone (MP) and an outer shell made of biocompatible polydopamine (PDA) (Figure 6A). Due to the adhesion of polydopamine, the prepared microcomposites (MP-PLGA@PDA) successfully reduced the excessive concentration of cytokines present after spinal cord injury. In addition, the controlled release of the immunosuppressant MP via microspheres inhibited the generation of inflammatory cascades. Therefore, the microcomposites inhibited the recruitment of macrophages and protected the injured spinal cord, thus leading to an improvement of motor function. In particular, this study deepens our understanding of the important role of neuroinflammation in neuroregeneration and functional repair. In recent years, various nerve guidance conduits have been developed as an alternative method for repairing peripheral nerve injuries. Liu et al. [137] prepared microspheres containing sustained-release nerve GF and implanted them into chitosan conduits to repair a defect of the buccal branch of the facial nerve in rabbits. These results indicated that the combination of chitosan nerve conduits and microspheres loaded with nerve growth factor could significantly promote a neural repair effect. 

Extracellular vesicles (EVs), an important category of microcapsules, can be used to deliver therapeutic cancer drugs such as microRNAs (miRNAs). Wang et al. [101] developed a microfluidic platform that could load therapeutic miRNA and drugs and control the size of microfluidically processed EVs (mpEVs) (Figure 6B). Because the delivery of therapeutic miRNAs through the BBB is limited, the intranasal injection of miRNA-loaded CXCR4-engineered mpEVs has been trialed in an orthotopic glioblastoma (GBM) mouse model, where the mpEVs were observed to travel through the nasal epithelia, bypass the BBB, and enter the intracranial region. This combination of packaging miRNAs in mpEV with non-invasive nasal administration has provided a new idea for the treatment of CNS diseases in clinical practice.

Hydrogels can be used as carriers of cells to create cell arrays for drug screening, monitoring intercellular interaction, mimicking ECM properties in a highly controlled manner, and also generating cellular microenvironments with varying properties [138,139]. Hydrogel microspheres are produced by droplet microfluidic technology, and the obtained hydrogel microspheres may be used as modular units and tightly stacked into a 3D porous structure. Hsu et al. [102] proposed an adaptable microporous hydrogel (AMH) for the fabrication of injectable scaffolds with interconnected micropores, which could direct and accelerate axonal regeneration in the presence of a propagating nerve growth factor (NGF)-gradient (Figure 6C). Such AHMs showed several advantages in overcoming the shortcomings of hydrogel-mediated tissue regeneration, including tunable mechanical properties, precise pore control, the effective induction of cell migration, and a propagating gradient of biological factors. Therefore, this unique hydrogel also provides a new approach for investigating the mechanism of neuroregeneration. In a previous study, Fe_3_O_4_@COOH nanoparticles were successfully combined with biotinylated dextran amine (BDA) to produce anterograde nano-neural tracers. These tracers were wrapped by microfluidic droplets to control leakage and allow for continuous and slow release. In addition, these multi-functional anterograde neural tracers had potential neural neurotherapeutic functions and could be used as a new platform for the integration of imaging and the treatment of peripheral nerve injuries [103]. Therefore, drug-delivery carriers engineered with microfluidic platforms have immense potential for future clinical applications.

## 4. Conclusions and Future Perspectives

Nerve injuries are common and frequently occurring diseases that can seriously affect the daily life activities and social function of patients. A major challenge in promoting functional recovery following injury and disorders of the CNS and PNS has been identifying the intrinsic mechanisms that orchestrate a regenerative response and enable axonal regeneration. Fortunately, modern microfluidic technology has shown significant potential for precisely controlling cellular behavior and the extracellular microenvironment; these factors are conducive to carrying out better research on neural degeneration and regeneration. Compared to in vivo animal models, in vitro models based on the microfluidics platform are rapid and efficient and are widely used in disease modeling. Different neural damage models, including those developed for traumatic injury and neurodegenerative diseases, can be built by using microfluidic technology and allow for the accurate investigation of the intrinsic mechanisms underlying nerve regeneration. Microfluidic platforms can also allow us to engineer the neural microenvironment to control and manipulate neurons or axons. This method can mimic the physiology of human organs and is highly reproducible and cost-effective. Engineered in vitro platforms hold immense potential for overcoming the many limitations of animal models, although in vitro models are unlikely to serve as the gold standard for neural damage research. It is widely believed that microfluidic platforms have substantial potential to enable us to further understand the specific mechanisms underlying neurological injury and disorders.

## Figures and Tables

**Figure 1 pharmaceutics-15-00210-f001:**
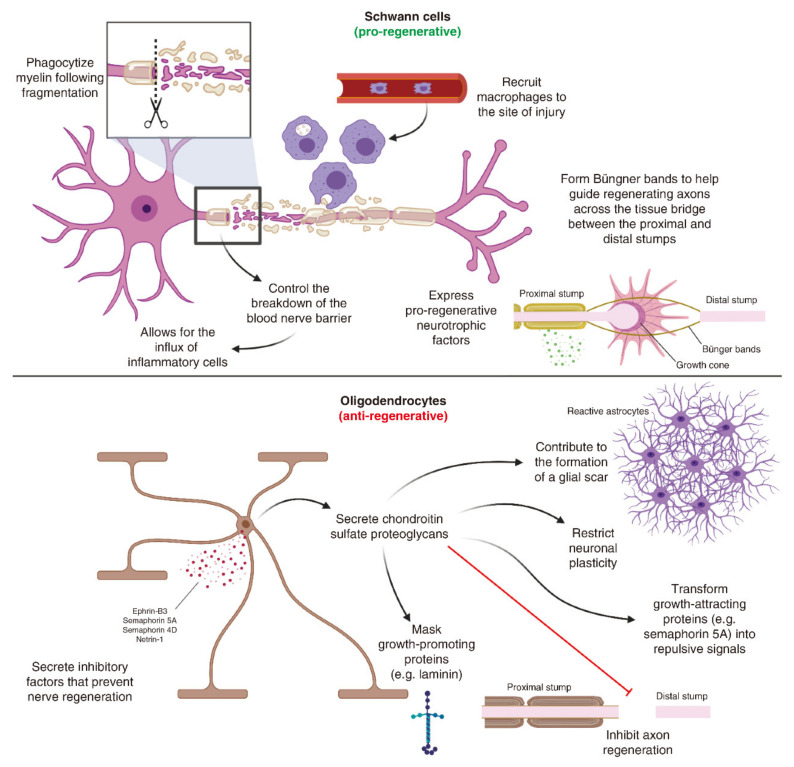
Comparing the role of Schwann cells and oligodendrocytes in nerve regeneration. Adapted with permission from Ref. [16]. Copyright 2021, Future Medicine Ltd.

**Figure 2 pharmaceutics-15-00210-f002:**
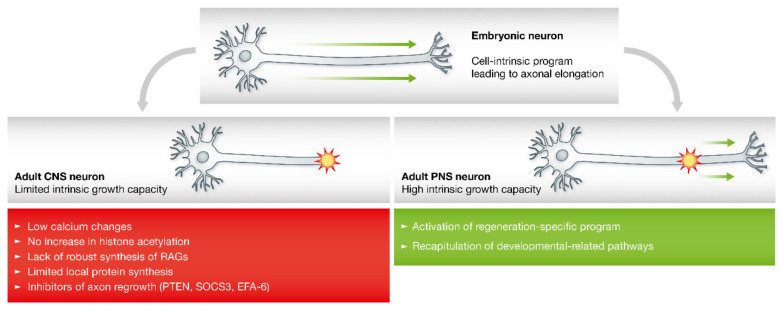
Increased growth capacity of PNS versus CNS neurons. CNS, central nervous system; EFA-6, exchange factor for ADP-ribosylation factor 6; PNS, peripheral nervous system; PTEN, phosphatase and tensin homolog; RAG, regeneration-associated gene; SOCS3, suppressor of cytokine signaling 3. Adapted with permission from Ref. [32]. Copyright 2014, Wiley-Blackwell.

**Figure 3 pharmaceutics-15-00210-f003:**
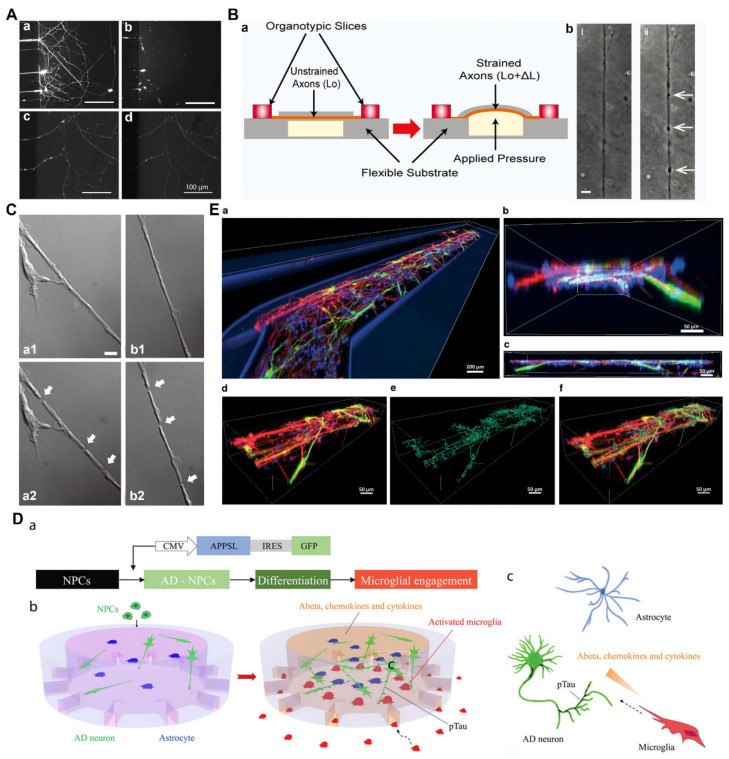
Engineering neural damage models using microfluidics. (**A**) Paclitaxel induced local axonal degeneration. Axons led to degeneration upon the axonal administration of paclitaxel for 24 h (**a**,**b**), but not with paclitaxel application to the soma chamber (**c**,**d**) ((**a**,**c**) = before paclitaxel; (**b**,**d**) = after paclitaxel). (**B**) A brain-on-a-chip to model mechanical axonal injury. Schematic of the uniaxial axonal strain device (**a**). Representative images of axonal beading (arrows) observed before (i) and after (ii) the strain injury (**b**). (**C**) A neuro-optical microfluidic device for precise axotomy. Reproducible spot damage (arrows) (**a1**,**a2**). Severity of spot damage using different laser energies (arrows) (**b1**,**b2**). (**D**) A 3D tri-culture microfluidic platform to model in vitro Alzheimer’s disease (AD). Schematic of the differentiation of neural progenitor cells (NPCs) to Alzheimer’s disease neurons and astrocytes (**a**). Schematics showing the multicellular interactions in the microfluidic AD model (**b**) and in the AD brain tissue (**c**). (**E**) A 3D microfluidic cell culture system to model in vitro Parkinson’s disease. Representative images of the 3D distribution of differentiated dopaminergic neurons. Top view of the entire culture chamber (**a**). Inside view (**b**) and side view (**c**) of a selected area. Top view of the selected area (**d**), reconstruction of the neuronal filaments of tyrosine hydroxylase (TH) and TUBβIII-positive neurons in the selected area (**e**), and overlap (**f**) of (**d**,**e**). (**A**) Adapted with permission from Ref. [82]. Copyright 2009, Elsevier; (**B**) Adapted with permission from Ref. [85]. Copyright 2014, World Scientific Publishing Co.; (**C**) Adapted with permission from Ref. [86]. Copyright 2009, the Royal of Society of Chemistry; (**D**) Adapted with permission from Ref. [88]. Copyright 2018, Nature; (**E**) Adapted with permission from Ref. [89]. Copyright 2015, the Royal of Society of Chemistry.

**Figure 4 pharmaceutics-15-00210-f004:**
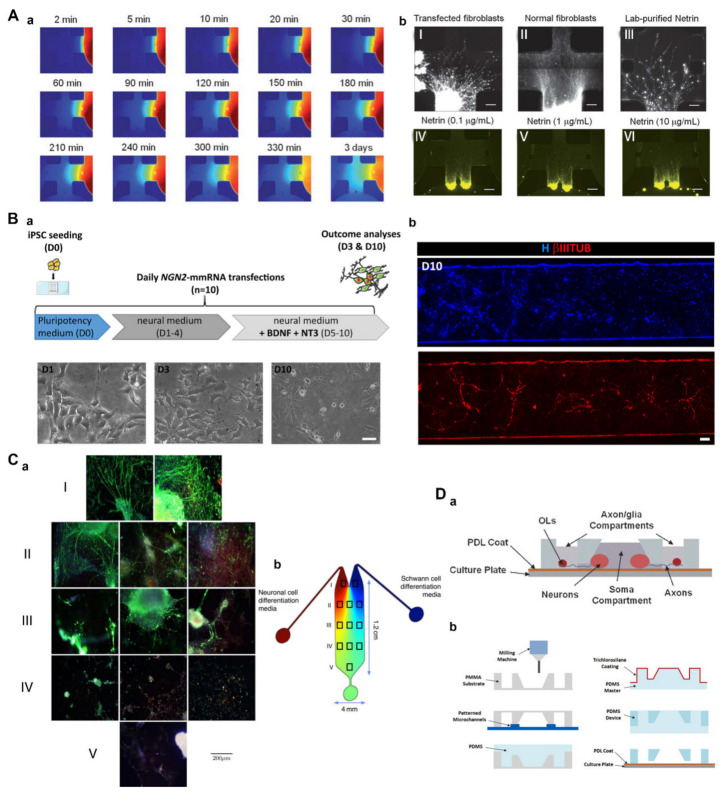
(**A**) A microfluidic device to study neurite guidance under chemogradients. Typical time-dependent snapshots of the gradient obtained with a 40 kDa FITC-dextran in the device (**a**). Neurite guidance of hippocampal neurons by chemoattractants (**b**). (**B**) NGN2 mRNA-based transcriptional programming. The NGN2-mediated transcriptional programming experiment of human-derived neural stem cells (hiPSCs) in microscale (**a**). Representative image of a single microfluidic channel of hiPSCs of NGN2 mmRNA transcriptional programming (**b**). (**C**) Differentiation of mouse embryonic stem cells (mESC) into neuron-like cells and Schwann cell-like cells in a microfluidic device. Photomicrographs (**a**) of the areas of the device, depicted as boxes in the cartoon (**b**). The top three rows (**I**–**III**) show significant neuronal differentiation and the directional outgrowth of neurites toward the “Schwann cell” sectors. The bottom two rows (**IV**,**V**) show no significant differentiation into any specific lineage. (**D**) A multi-compartment co-culture microfluidic platform. Schematic of the high-throughput microfluidic multi-compartment CNS neuron co-culture platform (**a**). Schematic of fabrication steps for the multi-compartment PDMS microfluidic device (**b**). (**A**) Adapted with permission from Ref. [91]. Copyright 2011, the Royal of Society of Chemistry; (**B**) Adapted with permission from Ref. [92]. Copyright 2021, Frontiers Media S.A.; (**C**) Adapted with permission from Ref. [94]. Copyright 2016, Wiley Periodicals, Inc.; (**D**) Adapted with permission from Ref. [95]. Copyright 2009, MYJoVE Corporation.

**Figure 5 pharmaceutics-15-00210-f005:**
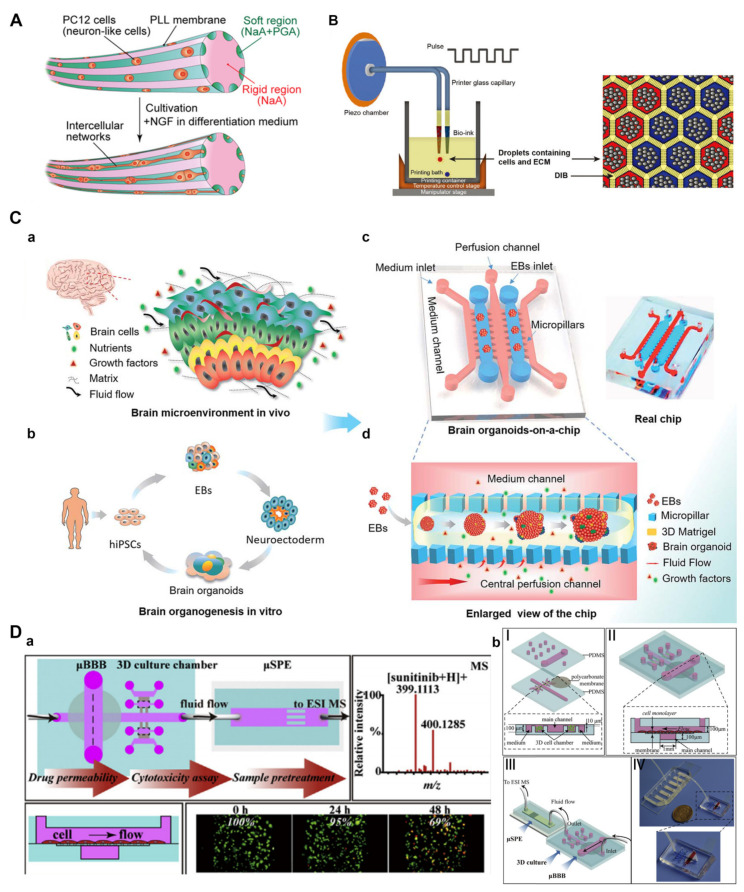
(**A**) Schematic of a complex hydrogel microfiber for guiding cell proliferation and forming intercellular networks. (**B**) The printed droplets connected by adhesive droplet interface bilayers (DIBs) using a lipid-bilayer-supported printing technique to form patterned droplet networks. (**C**) Schematic diagram of the brain organoids-on-a-chip device. The key factors of the brain microenvironment in vivo (**a**). The development process of brain organoids derived from hiPSCs in vitro (**b**). Configuration of the brain organoids-on-a-chip device (**c**). Enlarged view of the procedures for brain organoids generation on the chip (**d**). (**D**) A microfluidic blood–brain barrier model for evaluating drug permeability and cytotoxicity for central nervous system drug screening (**a**). Schematic illustration of the microfluidic platform (**b**). (**A**) Adapted with permission from Ref. [99]. Copyright 2014, IOP Publishing Ltd.; (**B**) Adapted with permission from Ref. [104]. Copyright 2020, WILEY-VCH Verlag GmbH & Co. KGaA, Weinheim; (**C**) Adapted with permission from Ref. [96]. Copyright 2018, the Royal of Society of Chemistry; (**D**) Adapted with permission from Ref. [98]. Copyright 2016, Elsevier.

**Figure 6 pharmaceutics-15-00210-f006:**
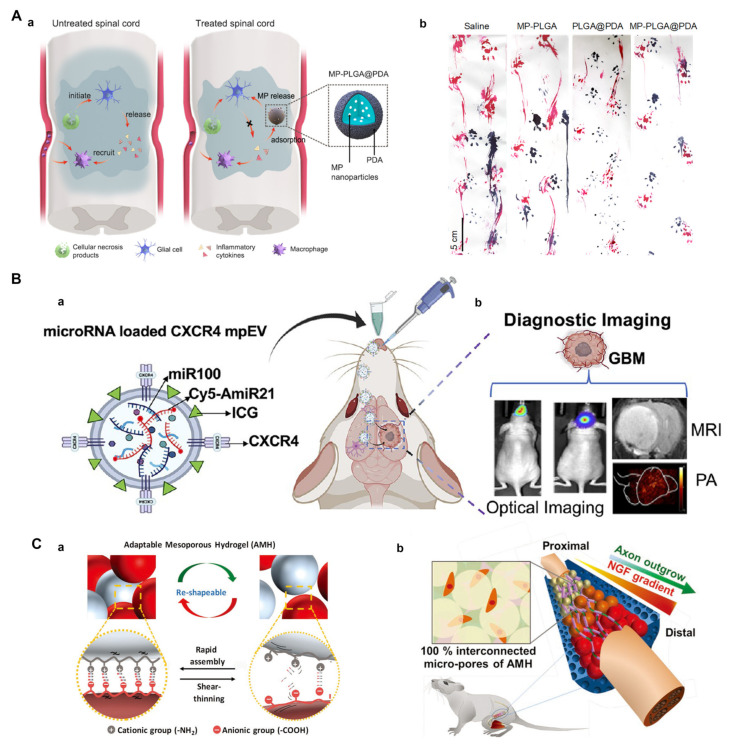
(**A**) Two-pronged cytokine suppressive strategy for promoting the functional recovery of the injured spinal cord by capturing the released cytokines and inhibiting the secretion of new ones (**a**). Representative footprint images of rats for assessing the recovery of motor function at day 28 post-injury (**b**). (**B**) Schematic of the intranasal administration of mpEVs in mice bearing orthotopic glioblastomas (GBMs) (**a**). The delivered mpEVs monitored using optical imaging, MRI, and photoacoustic imaging (**b**). (**C**) A sTable 3D scaffold formed by mixing oppositely charged building blocks via interaction (**a**). Schematic of adaptable microporous hydrogel (AMH) with the gradient propagation of NGF for directed and accelerated axonal regeneration in vivo (**b**). (**A**) Adapted with permission from Ref. [100]. Copyright 2021, American Chemical Society; (**B**) Adapted with permission from Ref. [101]. Copyright 2021, American Chemical Society; (**C**) Adapted with permission from Ref. [102]. Copyright 2019, WILEY-VCH Verlag GmbH & Co. KGaA, Weinheim.

**Table 1 pharmaceutics-15-00210-t001:** Summary of the different microfluidic devices and their applications to the nervous system. Abbreviations: 3D, three-dimensional; CNS, central nervous system; PNI, peripheral nerve injuries.

Type of Device	Application to the Nervous System	References
Polymer-based microfluidic devices	Chemical injury models of axons	[82,83]
	Mechanical injury models of axons	[84,85,86]
	Models of Alzheimer’s disease	[87,88]
	Models of Parkinson’s disease	[89]
	Models of amyotrophic lateral sclerosis	[90]
	Study of neurite guidance	[91]
	Inducing pluripotent stem cells toward neural fate	[92]
	Inducing the proliferation and differentiation of neural stem cells	[93]
	Induced stem cell-derived neuron-like cells and Schwann cell-like cells	[94]
	Study of the signaling networks of CNS	[95]
	Engineering stem cell-derived 3D brain organoids	[96]
	Brain permeability studies	[97,98]
Capillary-based microfluidic devices	Guiding network formation of neural cells	[99]
	Promoting the functional recovery of an injured spinal cord	[100]
	Enhanced therapeutic microRNA loading for CNS diseases	[101]
	Accelerating axonal regeneration	[102]
	Long-term tracking and repair of PNI	[103]
3D-printed devices	Constructing 3D tissue models and guiding self-organization	[104]

## Data Availability

Not applicable.

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
