# Peer review of "Microfluidic Manipulation for Biomedical Applications in the Central and Peripheral Nervous Systems"

_pharmaceutics, 2023, doi:10.3390/pharmaceutics15010210_

Round 1
Reviewer 1 Report
As mentioned in the text (abstract section), Li and colleagues aim at describing how the modern microfluidic platforms have impacted on high-throughput cell and organoid culture in vitro, the synthesis of a variety of tissue engineering scaffolds and drug carriers.
The choice of these hot topics in current science makes the review potentially interesting, and English language is fine, but the manuscript fails in keeping its premises. It merges several concepts and it is difficult to follow the logical flow of the text.
The first section is very long and dispersive. I would suggest to shorten it, and highlight the mechanisms that we are currently able to recapitulate with a microfluidic system.
The section presenting the different types of devices is poorly integrated with the rest of text. I would suggest to improve this part and explain what the advantages of the different fabrication techniques are (with reference to the first, more "biological", part).
The final part describes the use of microfluidic devices in the field of neural tissue and it sounds like a list of studies. I would suggest to highlight the advantages of each study with respect to the state of the art and our capability to recapitulate the mechanisms described in the first section.
All the images in the text are reproduced from other studies. I would like to encourage the Authors to prepare their own images and/or tables.
Finally, I would ask the Authors to write in vitro in italics.
Reviewer 2 Report
In their review entitled “Microfluidic Manipulation for Biomedical Applications in the 2 Central and Peripheral Nervous System”, the Authors describe different conditions of nerve system damage, including cell models for studying such damages. Most important, they summarize the advances done with microfluidic platforms, and their possible use for drug delivering in the context of neurodegeneration.
The paper is of interest and suitable for Pharmaceutics. Only minor points should be considered before acceptance:
1. 1. Although the paper already contains 6 very clear and well-drawn pictures, I suggest to add a table to summarize the different kinds of microfluidic approaches and their application to the Nervous System;
2. 2. Line 81: “tens of thousands of neurons and glial cell” is rather reductive when speaking about the human Nervous System, where it is better to speak about billions;
3. 3. Lines 421-423: not only exosomes but also microvesicles (MVs) are able to transfer different classes of molecules (mRNA, miRNAs, LncRNAs, proteins, metabolites, and so on); it is better, in this sense, to simply speak about extracellular vesicles.
Reviewer 3 Report
Introduction not well structured (in the first 47 lines). Fails to lead convincing to the topic of the review. Section 2 "Neurological Injury and Disorders of the Central and Peripheral Nervous Systems" is redundant. Reseachers who can benefit from this review already have knowledge of the central nervous system. Paragraph 3.1 and 3.2 are to genral, general fabrication methods of microfluidic system don't need to be discussed in a review on a speciic topic. Describtion of fabrication method need to have a closer connection to the topic.
Round 2
Reviewer 1 Report
I wish to thank the Authors for providing a new version of their manuscript. I would have preferred a more extensive revision, especially in the Introduction and in the parts focused on fabrication techniques, but they tried to reply to my comments. The manuscript is suitable for publication.
Author Response
Thank you very much for your suggestion. We believe the comments of you have significantly improved the quality and impact of the manuscript. Your comments are of guiding significance for our future writing. Thank you again for your sincere help.